# Towards Evaluating Proactive and Reactive Approaches on Reorganizing Human Resources in IoT-Based Smart Hospitals

**DOI:** 10.3390/s19173800

**Published:** 2019-09-02

**Authors:** Gabriel Souto Fischer, Rodrigo da Rosa Righi, Cristiano André da Costa, Guilherme Galante, Dalvan Griebler

**Affiliations:** 1Programa de Pós-Graduação em Computação Aplicada—PPGCA, Universidade do Vale do Rio dos Sinos—Unisinos, Av. Unisinos 950, Bairro Cristo Rei, São Leopoldo CEP 93022-750, Rio Grande do Sul, Brazil; 2Programa de Pós-Graduação em Ciência da Computação—PPGComp, Universidade Estadual do Oeste do Paraná–Unioeste, Rua Universitária 2069, Bairro Jardim Universitário, Cascavel CEP 85819-110, Paraná, Brazil; 3Parallel Applications Modeling Group—GMAP, Pontifical Catholic University of Rio Grande do Sul—PUCRS, Av. Ipiranga 6681, Bairro Partenon, Porto Alegre CEP 90619-900, Rio Grande do Sul, Brazil; 4Laboratory of Advanced Research on Cloud Computing–LARCC, Três de Maio Educational Society—SETREM, Av. Santa Rosa 2405, Três de Maio CEP 98910-000, Rio Grande do Sul, Brazil

**Keywords:** internet of things, health, smart hospitals, data prediction, sensors, distributed systems, human resources, elasticity

## Abstract

Hospitals play an important role on ensuring a proper treatment of human health. One of the problems to be faced is the increasingly overcrowded patients care queues, who end up waiting for longer times without proper treatment to their health problems. The allocation of health professionals in hospital environments is not able to adapt to the demands of patients. There are times when underused rooms have idle professionals, and overused rooms have fewer professionals than necessary. Previous works have not solved this problem since they focus on understanding the evolution of doctor supply and patient demand, as to better adjust one to the other. However, they have not proposed concrete solutions for that regarding techniques for better allocating available human resources. Moreover, elasticity is one of the most important features of cloud computing, referring to the ability to add or remove resources according to the needs of the application or service. Based on this background, we introduce Elastic allocation of human resources in Healthcare environments (ElHealth) an IoT-focused model able to monitor patient usage of hospital rooms and adapt these rooms for patients demand. Using reactive and proactive elasticity approaches, ElHealth identifies when a room will have a demand that exceeds the capacity of care, and proposes actions to move human resources to adapt to patient demand. Our main contribution is the definition of Human Resources IoT-based Elasticity (i.e., an extension of the concept of resource elasticity in Cloud Computing to manage the use of human resources in a healthcare environment, where health professionals are allocated and deallocated according to patient demand). Another contribution is a cost–benefit analysis for the use of reactive and predictive strategies on human resources reorganization. ElHealth was simulated on a hospital environment using data from a Brazilian polyclinic, and obtained promising results, decreasing the waiting time by up to 96.4% and 96.73% in reactive and proactive approaches, respectively.

## 1. Introduction

The Internet-of-Things (IoT) is a concept where physical objects (i.e., things) are connected through a network structure and are part of the internet activities in order to exchange information about themselves and about objects and things around themselves [1,2]. A particularly relevant scenario for IoT is healthcare [3,4,5]. IoT-assisted patients can be supervised uninterruptedly, thus allowing risky situations to be detected and appropriately treated right away [6]. According to Butean et al. [7], no matter how easy or complicated a situation is, if the medical staff do not react in an appropriate time, everything regarding patients’ health might become doubtful and unsafe. Hence, health professionals play a major role towards patients’ well-being [8]. In this kind of scenario, a static allocation of health professionals to health sectors may be inefficient, since some professionals may be misallocated to low demanding sectors, while leading to a lack of professionals in highly demanding sectors. Such a problem is illustrated in Figure 1, where the set of available attendants are statically assigned to two service sectors, one for exams and another for medication. In the example, more attendants are examining than medicating patients, even though the number of patients waiting for exams is considerably smaller than those waiting to receive some medication. In this context, if each room has a required specialty, and if each health professional has a list with all its specialties, the idle attendants who have the required destination room specialty could be moved from the low demanding room to the high demanding one. In fact, the allocation of attendants should always adapt to the current conditions of the health sectors.

Therefore, it is necessary to find effective strategies to adapt human resources in real-time. Elasticity in cloud computing is one of the key strategies for adapting on-demand computational resources [9,10,11]. According to Rostirolla et al. [12], the elasticity concept can be extended to other areas besides computing. Today, most resources control approaches can be classified as reactive or proactive (also named by some authors as predictive) [9,10,13,14]. Reactive approaches are based on both static bounds and if-condition-then rules to manage elasticity [9]. Typically, users define an upper and a lower threshold on a target performance metric (e.g., CPU utilization, memory, response time) to trigger activation and deactivation, respectively, of a certain resources number [15]. A problem of using fixed thresholds is related to application overloading, illustrated in Figure 2. After the system reaches an upper bound, there is a time interval for the delivery of the resource. During that period, we have an application overload [9]. Also, another problem is the lack of reactivity when using these parameters. There are situations in which is possible to anticipate the (de)allocation of resources, however, the resource configuration remains the same due to bad choices on setting the lower and upper thresholds [9,15].

A proactive approach employs prediction techniques to anticipate system behavior (its load) and thereby decide the adapting actions [9]. This capability enables the application to be ready to handle the increase when it actually occurs [15]. To accomplish this approach, it is common to use time-series-based prediction techniques (such as Exponential Smoothing, Moving Averages and Autoregressive models) and machine learning algorithms (including Neural Network, Linear Regression, Support Vector Machine, Reinforcement Learning and Pattern Matching techniques) [9,15]. This approach is typically classified adversely as time-consuming for sensitive performance applications [9,16]. Also, Netto et al. [17] affirm that proactive elasticity strategies focus on method accuracy and ignore limitations such as the scaling up operation time, although it dependents on the workload characteristics. Hence, the reactive approach performs faster because there is no concurrent processing concerning the application. In the proactive approach, for each monitoring step, it runs a given prediction algorithm that can impact the normal execution of the application, since the background task can be costly.

Considering this background, we present a model of Elastic allocation of human resources in Healthcare environments (ElHealth, for short) as an alternative to the traditional static allocation of medical staff. ElHealth works by adjusting the medical staff allocation of smart hospitals (equipped with IoT sensors) based on reactive and proactive elasticity approaches. In particular, ElHealth uses IoT sensors to keep track of patients demand, which is modeled as a time series and is used to estimate demands. Such estimations allow to identify situations where the staff availability is unlikely to meet the demand. Building upon such estimations, ElHealth proposes an efficient allocation of the medical staff by moving such professionals and also allocating new human resources to the most demanding areas while taking into account their time constraints. The idea is to always offer a reasonable waiting time for patients regardless of the workload (number of them in the hospital room). In resources elasticity, there are advantages and disadvantages in reactive and predictive methods. Using ElHealth, we propose an evaluation of proactive and reactive approaches for reorganizing human resources in smart hospitals to identify which significantly decreases the waiting time regarding healthcare. ElHealth supports both elasticity approaches at run-time. The main scientific contributions of this article are threefold:(i)We devise Human Resources IoT-based Elasticity, for automatic management of human resources in healthcare environments, making use of elasticity for smart, IoT-enabled hospitals;(ii)A cost-benefit analysis of the use of reactive and predictive strategies (of elasticity in cloud computing) for human resources reorganization. The cost refers to the health staff allocation costs in each approach, and the benefit is the anticipation of problems, based on the reduction of waiting time for care.(iii)We introduce Human resources cost and Elastic number of human resources used metrics for evaluating human resources elasticity.

This article is organized as follows. Section 2 presents the work related to our study. Section 3 presents ElHealth as well as the concepts of Multi-level Reactive and Proactive Elasticity of Human Resources. Section 4 expresses the methodology of evaluation of the model. Section 5 presents an evaluation performed with the developed implementation, as well as the results found. Finally, Section 6 presents the conclusions and future work directions.

## 2. Related Work

This section describes some approaches to manage elasticity in cloud and overviews approaches to managing the deficiency of resources to attend patients’ demand in healthcare environments. They were divided into two groups: reactive and proactive systems in Section 2.1 (where we discuss two papers of elasticity in cloud computing, one for each approach, and all articles found that extend the concept of elasticity to other areas) and human resources in Section 2.2 (where we discuss some works that focus on human resources lack in healthcare environments). Lastly, the initiatives were compared and analyzed in order to detach the current gaps in the research area.

### 2.1. Reactive and Proactive Systems

Reactive managers are those based only on thresholds to take elasticity decisions; more precisely, resource reconfiguration takes place when the lower or the upper threshold is violated. In the reactive scope, we highlight three initiatives: Al-Dhuraibi et al. [18], Elastic-RAN [19] and ElCity  [12]. Al-Dhuraibi et al. [18] presents a new elasticity management system powering both vertical and horizontal elasticities, both VM and Container virtualization technologies, multiple cloud providers simultaneously, and various elasticity policies based on a dynamic configuration during the execution of the application. The experiments demonstrated that their model covers the elasticity policies provided by the well-known cloud public providers with negligible overhead. Elastic-RAN [19] proposes a multi-level and adaptable elasticity for Cloud Radio Access Networks (C-RANs). The adaptive algorithm feature refers to the moldable elasticity grain where resources in BBU pools level and BBU level are provisioned as close as possible to the current processing needs. Elastic-RAN might achieve gains up to 64% in the execution time when compared to a traditional C-RAN. ElCity [12] is a model that combines citizens and city devices data to enable an automatic and elastic multi-level management of energy consumption for a particular city. ElCity explores the cloud elasticity concept in multiple target levels (smartphones from citizens, city devices involved in the public lighting, and data center nodes), turning on or off the target levels resources on each level regarding their demands, estimated based on energy consumption monitoring and citizens movement. ElCity achieved a reduction of more than 90 percent of the energy spent in public lightning in the studied city.

Proactive managers try to predict the cloud behavior to anticipate elasticity decisions before any under or overload situation. In the proactive elasticity, we highlight two works: Hanafy et al. [20] and Proliot [21]. Hanafy et al. [20] proposed an elasticity control algorithm for a containerized cloud using two agents. The host agent monitors and predicts its utilization using Autoregressive Moving Average (ARMA) [22], while the master agent performs elasticity by handling failures in load interchange scenarios. The results demonstrated the algorithm capabilities to elasticate and handle flash crowds along with decreasing the management overhead and maintaining proximate load balancing. Proliot [21] combines cloud and high-performance computing to address the IoT scalability problem in a novel EPCglobal-compliant architecture. The model offers an elastic EPCIS component that is automatically allocated or deallocated concerning the system load. Proliot uses Autoregressive Integrated Moving Average (ARIMA) [23] and Weighted Moving Average (WMA) [24] to predict the IoT load behavior, anticipating scaling in or out operations. Proliot improves 300% the response time when compared with the scenario that is not using elasticity. Table 1 summarizes the aforementioned related work. Reactive approaches have a low computational cost compared to proactive approaches. However, proactive approaches can avoid overloading in applications by taking elasticity actions in advance.

### 2.2. Human Resources in Healthcare Environments

Some approaches focused on optimizing the flow of patients to properly allocate health resources [25,26,27]. Cappoci et al. [25] used discrete event simulation technique in order to improve patients’ waiting times. To this end, using data from a Brazilian polyclinic, and queueing theory [28], the authors proposed some changes to balance the occupancy levels of the health unit’s staff and, at the same time, reach a shorter waiting time for patients. Results showed a significant improvement in the performance of the Polyclinic’s system. Vieira and Hollmén [26] investigated ways of minimizing bottlenecks in the flow of patients due to appointments, visits, usage of resources, etc. The objective was to improve patients’ satisfaction and maximize the hospital’s profit. To this end, using data from a Finnish hospital, the authors used k-Nearest Neighbours [29,30] and Random Forests [31] to predict such a flow. In the same line of thinking, Graham et al. [27] aimed at predicting the arrival of patients in the emergency department of a hospital to properly prepare the allocation of medical staff. To accomplish such a task, the authors used logistic regression [32], decision trees [33], and gradient boosted machines [34] with data from a British hospital. In both works [26,27], the objective was exclusively on identifying specific data patterns, instead of proposing counter-measures to improve the allocation of health resources.

In an attempt to increase health coverage, some studies proposed forecasting models to understand the evolution of doctors supply and patients demand to better adjust one to the other. Ishikawa et al. [35] concentrated on training enough physicians to meed the patients demand in Japan until 2030. Liu et al. [36] focused on a similar problem, but from a global perspective. In contrast to our work, the adaptation of the hospital’s resources to the patients’ flow was left aside for these works. Table 2 summarizes the aforementioned human resources related work. As we can see, there are several approaches to analyze and estimate the use of human resources in healthcare environments so that that patient flow can be improved, or to understand the evolution of the problem of the health professionals lack.

### 2.3. Comparison and Research Opportunities

Table 1 and Table 2 presents a comparison of the collected papers, presenting some of their main characteristics, and pointing out some of their gaps. Based on the selected papers, we can identify that despite the elasticity being proposed for cloud computing, and being employed in reactive [37] and proactive [20,38] approaches, the same can also be employed in other areas such as energy [12], IoT [21] and C-RAN [19]. In this way, we can see the potential of elasticity to be extended to other contexts, such as human resources. When we have the problem of the lack of resources in hospital environments, the articles found just focus on predicting the future demand of patients or the future quantity of available doctors, not proposing solutions to the problem, leaving others in charge of decision-making. The approaches that propose solutions, such as physician training [35], or the movement of a nurse between two rooms [25], are very specific and can not be used in other medical environments. In this context, we can enumerate some of the main gaps in the area as follows:In the best of our knowledge, there are no approaches that evaluate the use of reactive and predictive elasticity for human resource management;Although several models are capable of identifying current and future demand in a hospital environment, these models lack *solutions* to help to solve the problem of deficiency of hospital resources;

The lack of enough human resources in healthcare environments is not new and, based on studied works, we can see that this problem will remain in the future [35,36]. Hence, finding ways of optimizing the use of existing resources and adjust hospitals’ capacity to meet patients demand are challenges that can make all the difference. The use of data prediction and Internet of Things contributes towards future solutions or automation of processes in the health area. However, the potential of the technologies is being underused since it is possible to propose solutions such as optimization and better use of existing human resources.

## 3. ElHealth Model

According to the literature review, most of the approaches concentrate only on identifying the location and current/future health status of patients, neglecting the potential benefits that efficient health resources allocation could bring to the patients [39,40]. As presented in Section 1, one of the major challenges faced in hospital environments refers to the large waiting queues. Moreover, considering that doctors reaction time plays a role in patients recovery [7], long waiting times may compromise patients’ future health.

Based on this background, we introduce ElHealth, a multi-level model for efficient allocation of human resources based on patients’ flow within hospital environments. In particular, ElHealth adapts the concept of elasticity in cloud computing to the context of human resources, adjusting the hospital’s attendance capacity to the demand of patients, where professionals are allocated, deallocated and reallocated according to the hospital needs. ElHealth groups information from several sources: patients arrivals and needs (using IoT sensors spread over the hospital environment and a hospital dataset), patients movement (using IoT sensors), and medical staff availability (from a dataset). Using these data, we measure real-time demand of patients, on reactive approach (which we discuss in Section 3.3.1), and we employ a time-series prediction algorithm to anticipate the future demand of patients, on proactive approach (as we discuss in Section 3.3.2). This information is then useful for applying the concept of elasticity-based allocation of resources. Based on that model, ElHealth computes an efficient allocation of hospital resources (medical staff and equipment), which contributes towards minimizing patients’ waiting queues. Hence, ElHealth introduces the concept of Human Resources IoT-based Elasticity in healthcare environments, which can be defined as follows.

**Definition** **1** (Human Resources IoT-based Elasticity)**.**
*Human Resources IoT-based Elasticity is an extension of the concept of resource elasticity in Cloud Computing [13] to manage the use of human resources in a healthcare environment, where health professionals are allocated and deallocated according to patients’ demand. The Human Resources IoT-based Elasticity uses IoT sensors to keep track of patients’ demand and, based on proactive and reactive elasticity approaches, proposes an efficient allocation of the medical staff by moving such professionals to the most demanding areas, always considering the quality of services currently offered by these healthcare environments.*


The next subsections detail our model, bringing the main design decisions (Section 3.1), the proposed architecture (Section 3.2), and the Multi-level Elasticity of Human Resources concept using reactive (Section 3.3.1) and proactive (Section 3.3.2) approaches.

### 3.1. Design Decisions

We based our model on the premise that there are sensors scattered around the hospital, which can identify patients who pass through them. Firstly, they must be in all the entrances and exits, so that whenever a patient enters or leaves the hospital, it is possible to identify it. To detect the movement and location of patients, we assume the presence of sensors at the doors of all hospital rooms. Each patient must have a Patient Identification Wristband linked in the system and must carry it through all time in the hospital’s internal environment. The attendant responsible for the reception of patients should be able to perform the linking of a wristband to a given patient as soon as the patient is admitted in the hospital. Thus it is possible to identify when and where a given patient is as soon as he enters at the healthcare environment, along with the time he remains in each room while being attended to. Also, each healthcare professional must have a tag linked to him in the system and must carry it with him throughout his active period in the hospital. Thus, all available attendants can also be located inside the hospital in the same way as patients.

We use a Real-Time Location System (RTLS) [41] with room-level localization accuracy. According to Boulos and Berry [41] and Jachimczyk et al. [42], RTLS are systems for identifying and tracking the location of assets and/or people in real-time or near real-time. Furthermore, RTLS provides an automated means of collecting operational data on clinic activity such as room utilization rates, or patient wait times [43]. We based the choice of an RTLS on its ability to allow automatic identification, avoiding the existence of a human error in identification processes. ElHealth should be transparent to patients, in the sense that it does not need to report any conditions related to its movement through the hospital environment, being an activity performed automatically by the system.

With respect to the data prediction strategy, ElHealth uses a statistical-based approach through the implementation of the ARIMA model. According to Nisha and Sreekumar [44], ARIMA model uses historical information to predict future patterns. ARIMA is the most general class of model for forecasting a time series. Since we can describe the number of patients waiting for care over time as a time series, we chose to use the approach through ARIMA because it is a very flexible mathematical model, with an excellent predictive performance of time series when compared with other approaches [44]. ARIMA models are extremely useful in predicting different sectorial series since they can represent stationary series, and also non-stationary series. We use a non-stationary model based on seasonality in demand for medical staff, since accidents, epidemics, holidays, and other events, can alter patients’ demand for care.

### 3.2. Architecture

ElHealth architecture model three services: (i) a Web service, responsible for visualization layer, and ElHealth Web Interface; (ii) an inference service, responsible for data processing, movement records handling, patients demand prediction, and human resources allocation decisions; and (iii) a database service. These three services are part of our proposed ElHealth Service. Figure 3 presents the components and the network view in the proposed model.

ElHealth model is subdivided into five modules responsible for information handling from its capture by sensors to the final result displayed in the Web application. Each module has a specific function, having an input information and a specific output result that can be used as input from other modules. Figure 4 presents the proposed modules, detailing the architecture of the model.

*ElHealth_Capture* receives and pre-process data captured by sensors scattered around the hospital and sends to *ElHealth_Formatter*, responsible for process data, and identify patients’ movement through hospital environments and rooms. After, *ElHealth_Predict* identifies patients movement through the hospital environment. Based on previously generated movement records, the path that patients travel during their movement through the hospital, and the time spent in each environment are identified. Thus, this module identifies patterns related to the arrival of patients in these environments, and patterns related to the waiting time for care, using this information to predict future patients arrivals in each hospital environment.

*ElHealth_Elastic* manages system’s elasticity. It verifies human resources allocation in each of hospital environments, check the current patients’ movement (in reactive approach) and the predictions made by the previous module (on proactive approach). This module generates an intelligent and automatic allocation of human resources to meet patient demand better. We want to emphasize that the system generates notifications for human resources to reallocate, but effective reallocation depends on the people accomplishing what was indicated by the application. *ElHealth_Elastic* and *ElHealth_Predict* modules are the most important part and the core of our proposed model, since ElHealth_Elastic can request predictions from the ElHealth_Predict module to take elastic actions, performing resources analysis based on predictions performed by the previous module, and also can perform elastic actions based on current patient demand. In Section 3.3 will be detailed the algorithms and how the elastic management of the human resources in the hospital environment are performed. Finally, *ElHealth Web Interface* displays to human resources the elasticity notifications generated before.

### 3.3. Human Resources Elasticity

ElHealth employs the term elasticity with a slightly different meaning from that used in cloud computing. Here, it refers to the system’s ability to allocate/reallocate/deallocate human resources capable of attending patients in order to adapt to varying patient demand in real-time. In particular, in the context of this work, elasticity refers to:**Allocation**, which denotes the capacity of the system to request health professionals who are not in the hospital to attend the current patients’ demand;**Reallocation (or migration)**, which denotes the ability of the system to migrate professionals who are allocated to a particular hospital environment to some other environment where more professionals are needed;**Deallocation** which denotes the capacity of the system to release human resources no longer needed to attend the current patients’ demand.

In order to perform allocation, deallocation, and reallocation of human resources, ElHealth model makes use of reactive or proactive approaches to monitor the demand of patients and the use of rooms in the hospital. Our model considers elasticity differently for: (i) the reactive approach, where our model must verify the use of any given room, and propose human resources movement if an upper or lower threshold is reached (as discussed next, in Section 3.3.1), and for (ii) the proactive approach, where ElHealth should verify if there are sufficient attendants to meet patients’ future demand from any given room in the hospital environment, with attendants moving between rooms (as detailed forward in Section 3.3.2). For this process, ElHealth should be able to alert people to allocate. However, the final decision should always be made by the health professional or hospital manager.

#### 3.3.1. Reactive Elasticity

In reactive mode, ElHealth uses a multi-level approach where our model considers elasticity differently at (i) the room-level, where ElHealth must verify the use of a given room, and check if is necessary more or fewer attendants to meet patients’ demand, and at (ii) the hospital-level, where our model proposes attendants movement between rooms to meet patients’ demand. We use this multi-level strategy, since different rooms may have different time thresholds for care. In this way, a prior analysis of the need for each room-level is necessary in order to perform the load-balancing procedure (hospital-level). An example of these two levels is presented in Figure 5.

ElHealth model adapts the reactive elasticity strategy using upper and lower thresholds for the context of people, based on the waiting time for care in each of waiting queues of a hospital environment. Figure 6 illustrates the use of thresholds where an upper threshold is reached (meaning that human resources should be increased to fulfill that needs) and soon after a lower threshold is reached (meaning that human resources could be released to other sectors). So, at room-level, in each monitoring cycle, ElHealth first checks the specific time thresholds of each analyzed room and compares with the waiting time in that room. In those where time is outside the upper or lower bounds, our model defines the need for allocation or deallocation of human resources.

At the hospital-level, ElHealth considers the possibility of moving health professionals between different hospital environments in order to optimize medical care time. To this end, the available options refer to: allocating new attendants, reallocating health professionals between different sectors, or deallocating human resources that are no longer necessary. ElHealth’s first option should always be the possibility of reallocating human resources already allocated to hospital care. The reallocation is prioritized because it is the option that brings fewer costs to the hospital since it performs adjustment of medical care without additional attendants. Algorithm 1 presents the pseudo-code for hospital-level reactive elasticity.

**Algorithm 1:** Hospital-Level Reactive Elasticity.
      **Data:** Hospital room list *h*, vector *v* with all attendants of hospital
      **Result:** Updated hospital room list *h*
  1 **begin**
  2   l← a new vector of rooms and quantity of attendants to allocate or deallocate;
  3   **forall**
*Room r on hospital room list h*
**do**
  4    q← execute *Room-level Reactive Elasticity Algorithm* using *r* as Data;
  5    l.add(r,q);
  6   **end**
  7   sort *l*, available attendants;
  8   l← execute for *Human Resources Deallocation Algorithm* using *l* and allocated attendants of *v* as Data;
  9   sort *l*, available attendants;
10   **forall**
*Room r on list l*
**do**
11    $l_{r}\leftarrow$ sort *l*, available attendants with room *r* specialty;
12    $available_{r}\leftarrow$ list of all human resources available for allocation with room *r* specialty;
13    execute *Human Resources Reallocation Algorithm* using *r* and $l_{r}$ as Data;
14    **if**
*r need more attendants*
**then**
15     Execute *Human Resources Allocation Algorithm* using *r*, $l_{r}$ and $available_{r}$ as Data;
16    **end**
17   **end**
18   h← rooms of *l* vector;
19   **return**
*h*;
20 **end**


In what follows, we firstly discuss the reallocation concept, followed by the allocation procedures, rules, and algorithms. Lastly, we present the deallocation process. We note that, although deallocation appears first in the algorithm (line 9), it actually builds upon the human resources allocated during the preceding iteration of the algorithm. In ElHealth model, each room has a required specialty to the human resources that are allocated in it. In parallel, each health professional has a list of all its specialties. The process of reallocating or allocating human resources is only performed between professionals who have the required destination room specialty. This is necessary because in a laboratory exams room is required a nursing professional accustomed to blood tests for example, and even if we have X-ray technicians available for reallocation, they are not able to improve the attendance in the aforementioned room. In order to achieve human resources reallocation, all hospital rooms are in a list ordered by the attendants available for reallocation. In that way, whenever a room *r* needs a new human resource, the elasticity manager checks for available attendants, with room *r* specialty, in the first room of the list. If there is an available attendant, then it is reallocated to the needed room.

A potential problem that arises in the context of elasticity is the so-called hysteresis [45], which refers to the tendency of the system to return to the previous state in the absence of the impulse that caused the change. In the context of human resources elasticity, hysteresis occurs if a resource reallocated from a given room A to another room B and, in the subsequent time-step, room A needs that resource back. This kind of situation happens when the stimulus that led to the reallocation ceases to exist. However, when the resource is returned to the original room, the stimulus will emerge once again, leading the resource to be reallocated continuously between the two rooms. In order to prevent hysteresis of human resources, we employ a cooldown-based strategy [46]. In particular, whenever a resource is reallocated from a given room A to another room B, and if room B need a resource in the subsequent monitoring cycle, its need will only be met if another room has free resources, or by the allocation of a new attendant. In other words, the resource reallocated previously cannot be immediately returned, which avoids the hysteresis effect.

In some situations, the reallocation process may not be enough to improve the attendance level of the hospital. In such situations, the allocation of new resources may be necessary. We emphasize that, in order to minimize operational costs, the allocation is only performed if reallocation is not able to meet the patients’ demand. In an emergency situation, or exceptional cases, where all hospital staff are already in care and not available for reallocation, ElHealth proposes the allocation of new human resources. Thus, our model allocates health professionals who are not in the hospital but are available for allocation. We highlight that the hospital must have a strategy to define human resources available for external allocation. Since different countries have different labor laws, the rules that can make available for allocation the hospital staff on rest time can vary. Finally, if the algorithm identifies that the demand for care of all hospital rooms is very low and that the deallocation of attendants of some room does not harm the whole, ElHealth must identify which attendants were allocated outside of their regular working hours and deallocate them to lower the hospital’s financial costs. In the same way as reallocation, both allocation and deallocation are also protected by the cooldown-period. Also, if a given human resource is deallocated, it can no longer be allocated in the same work shift.

#### 3.3.2. Proactive Elasticity

In proactive elasticity, ElHealth model uses a multi-level approach, slightly different of reactive elasticity, where (i) in the room-level, our model must identify the future use of a given room, and check if the number of attendants is sufficient to meet patients’ demand, and in (ii) the hospital-level, where ElHealth should verify if there are sufficient attendants to meet patients’ demand from all rooms in the hospital environment, with attendants moving between rooms. An example of these two levels is presented in Figure 7.

ElHealth model adapts the proactive elasticity strategy using upper and lower thresholds for the context of people, based on the waiting time for care in each of waiting queues of a hospital environment. Figure 8 illustrates the use of thresholds, where ElHealth forecasts that the upper threshold will be reached and soon after ElHealth forecasts that the lower threshold will be reached.

At the room-level, in each monitoring cycle, ElHealth needs to predict patients arrival rate at any room based on current and previous arrivals on that room. The prediction is made using the ARIMA model based on the average care time with the current attendants’ allocation, and the estimated waiting time for the care queue. When ElHealth identifies that the waiting time will become higher or lower than the threshold values set by hospital manager, ElHealth should compute the number of health resources required to meet patients’ demand through the Proactive Human Resources Elastic Speedup. Proactive Elastic Speedup uses a predictive approach to determine the future demand of patients and dynamically define the adequate number of attendants, identifying the gain of future medical care time in a hospital environment. ElHealth proposes some mathematical formalism to estimate the Proactive Human Resources Elastic Speedup, which will be described in the sequence. Table 3 presents a summary of such mathematical notation.

Let $CV(r,t_{i},t_{f})$ denote the care vector of room *r* for the time interval between $t_{i}$ and $t_{f}$. The size of any such vector is defined by size(x). Using these two functions, the average care time in the hospital’s room *r* between $t_{i}$ and $t_{f}$ times can be formulated as in Equation (Equation 1), where CDT(x[i]) refers to a care duration time x[i] that has already occurred in that room and $x\left[\right]=CV\left(r,t_{i},t_{f}\right)$ is a care vector that occurred in that room.

(1)$$\begin{array}{c} ACT\left(r,t_{i},t_{f}\right)=\frac{1}{size(x)}\sum_{i=0}^{size(x)-1}CDT\left(x\left[i\right]\right) \end{array}$$

Equation (Equation 1) results in a numerical value of time. An example would be any room *r*, between 1 and 5 times, where the result could be defined as: ACT(r,1,5)=15 minutes. Using this equation, it is possible to estimate the average time of a care in a particular hospital room. Due to the elasticity of human resources, at different time instants, there is a different number of attendants allocated to care in each of the hospital rooms. The average number of attendants in the hospital’s room *r* between times $t_{i}$ and $t_{f}$ is defined as in Equation (Equation 2), where $NA(r,t_{n})$ refers to the number of attendants allocated to care in the room *r* at the instant of time *n*.

(2)$$ANA\left(r,t_{i},t_{f}\right)=\frac{1}{t_{f}-t_{i}}\sum_{t_{n}=t_{i}}^{t_{f}-1}NA\left(r,t_{n}\right)$$

The same idea of the previous function is useful for patients’ reality because in different moments of time there are different amounts of patients awaiting care in each of the hospital rooms. Thus, the estimated number of patients waiting for care in the hospital’s room *r* between $t_{i}$ and $t_{f}$ times is defined by Equation (Equation 3), where $NWP(r,t_{i})$ refers to the number of waiting patients for care in a room *r* at $t_{i}$ time instant, and $NIP(r,t_{n})$ refers to the number of incoming patients in a room *r* at $t_{n}$ time instant.

(3)$$ENP\left(r,t_{i},t_{f}\right)=NWP\left(r,t_{i}\right)+\sum_{t_{n}=t_{i}+1}^{t_{f}-1}NIP\left(r,t_{n}\right)$$

Using the equations previously proposed, our model calculates the estimated care time of all patients waiting, and estimates the time that a new incoming patient needs to wait to be attended. The $ECT(r,t_{i},t_{f})$ is defined by Equation (Equation 4), where $ACT(r,t_{i},t_{f})$ refers to the average care time for room *r* between $t_{i}$ and $t_{f}$ times, and $ENP(r,t_{i},t_{f})$ refers to the estimated number of patients who are waiting for care in a room *r* between $t_{i}$ and $t_{f}$ instants. An example would be the room $r_{1}$, between two times $t_{i}$ and $t_{f}$ that would result in an average number of 4 patients and an average care time of 10 min as shown in Figure 9.
(4)$$ECT\left(r,t_{i},t_{f}\right)=ACT\left(r,t_{i},t_{f}\right)\cdot ENP\left(r,t_{i},t_{f}\right)$$

Knowing $ECT(r,t_{i},t_{f})$, we can analyze the average time for care of all patients waiting in the room *r* between $t_{i}$ and $t_{f}$ times. However, this value refers to a hospital room with a single attendant allocated for care, but in most cases will be more than one health professional working in that room, making it necessary to identify the average time with different numbers of attendants. In this context, ElHealth model uses a parallel allocation of human resources, such as the parallel allocation of virtual machines used in elastic systems [13] or the use of parallel processors in high-performance computing [47]. Thus, based on the Elastic Speedup proposed by [47], ElHealth introduces Equation (Equation 5) for Human Resources Elastic Speedup. Considering again the previous example (Figure 9), with room $r_{1}$ between two times $t_{i}$ and $t_{f}$ with an average number of 4 patients, an average care time of 10 min and with two health professionals allocated, as shown in Figure 10.
(5)$$HRES\left(r,t_{i},t_{f}\right)=\frac{ECT(r,t_{i},t_{f})}{ANA(r,t_{i},t_{f})}$$

$HRES(r,t_{i},t_{f})$ returns the estimated care time of a room *r* between the $t_{i}$ and $t_{f}$ times, considering a parallel allocation of attendants in that period of time, through the use of $ANA(r,t_{i},t_{f})$ function. Thus, with the increase in the average number of attendants allocated, the estimated care time decreases, inversely proportional.

A problem of reactive elasticity is that the elasticity actions are taken after the upper threshold are reached, causing a state of overload in the hospital throughout the professionals’ movement period. Thus, an alternative to this problem is the use of proactive elasticity [48]. Thus, anticipating the moment when the upper threshold will be reached, people’s movement can occur in advance, minimizing or avoiding patients’ overloads in the hospital. In this context, we propose Equation (Equation 6) for Proactive Human Resources Elastic Speedup as follows:(6)$$PHRES\left(r,a,f_{i},f_{f}\right)=\frac{ECT{\left(r,f_{i},f_{f}\right)}^{\prime }}{a},$$
where *a* is the number of attendants allocated between the future times $f_{i}$ and $f_{f}$, and $ECT{\left(r,f_{i},f_{f}\right)}^{\prime }$ is a prediction of the future care time for this room using ARIMA. We can compute $ECT^{\prime }$ as:$$ECT{\left(r,f_{i},f_{f}\right)}^{\prime }=ACT{\left(r,f_{i},f_{f}\right)}^{\prime }\cdot ENP{\left(r,f_{i},f_{f}\right)}^{\prime },$$
where $ACT{\left(r,f_{i},f_{f}\right)}^{\prime }$ and $ENP{\left(r,f_{i},f_{f}\right)}^{\prime }$ are predictions of the average care time and future patients at room *r*, respectively. Thus, for each room *r* being calculated, we generate a time series of $ACT(r,t_{i},t_{f})$ that occurred in the past, and we use it to predict $ACT{\left(r,f_{i},f_{f}\right)}^{\prime }$. In addition, for each room we also generate a time series for $NIP(r,t_{i},t_{f})$, and can predict future patient input and find $ENP{\left(r,f_{i},f_{f}\right)}^{\prime }$.

Using the aforementioned equations, ElHealth can predict the waiting time of any hospital room. Varying *a* attribute in PHRES equation, with the increase and decrease of the number of health professionals in attendance, ElHealth can identify how many attendants would be needed to adjust the waiting time of any room to the proposed thresholds, as defined by the hospital manager. Algorithm 2 presents our method to verify the need to allocate or deallocate human resources in any room *r* in a smart hospital.

**Algorithm 2:** Room-Level Predictive Elasticity.
      **Data:** Room *r*, *a* attendants, future initial time $f_{i}$, future final time $f_{f}$
      **Result:** Quantity of attendants to allocate or deallocate
  1 **begin**
  2   upper← Upper Threshold of waiting time in *r*;
  3   lower← Lower Threshold of waiting time in *r*;
  4   n←0;
  5   $a^{\prime }\leftarrow a$;
  6   **if**
$PHRES(r,a,f_{i},f_{f})>upper$
**then**
  7    **while**
*$a^{\prime }<limit\left(r\right)$ e $PHRES(r,a^{\prime },f_{i},f_{f})>upper$*
**do**
  8     n←n+1;
  9     $a^{\prime }\leftarrow a+n$;
10    **end**
11   **else if**
$PHRES(r,a,f_{i},f_{f})<lower$
**then**
12    **while**
*$a^{\prime }>0$ e $PHRES(r,a^{\prime },f_{i},f_{f})<lower$*
**do**
13     n←n−1;
14     $a^{\prime }\leftarrow a+n$;
15    **end**
16   **end**
17   **return**
*n*;
18 **end**


At the hospital-level, ElHealth needs to test different allocations for the attendants to ensure that all rooms identified in the previous step (local-level) have enough attendants, and to minimize overcrowding. Our algorithm considers the possibility of moving health professionals between different hospital environments in order to optimize medical care time. As in the reactive strategy, between allocation or reallocation, ElHealth prioritizes the possibility of reallocating human resources already allocated to hospital care, to minimize hospital’s costs. To redistribute such health attendants between different hospital rooms, our model uses some strategies known from other contexts of scientific computing and adapts them to the proactive elasticity of human resources needs. Algorithm 3 presents the pseudo-code for hospital-level proactive elasticity. As in the reactive strategy, each room has a required specialty, and the process of reallocating or allocating human resources is only performed between professionals who have the required destination room specialty. A point to be observed is that those rooms where they need a specialty that no other hospital’s professional has, only the allocation of new human resources is performed.

**Algorithm 3:** Hospital-Level Predictive Elasticity.
      **Data:** Hospital room list *h*, vector *v* with all attendants of hospital, future initial time $f_{i}$, future final time $f_{f}$
      **Result:** Updated hospital room list *h*
  1 **begin**
  2   l← a new vector of rooms and quantity of attendants to allocate or deallocate;
  3   **forall**
*Room r on hospital room list h*
**do**
  4    a← number of attendants allocated in *r*;
  5    q← run Algorithm 2 for *Room-level Predictive Elasticity* using *r*, *a*, $f_{i}$ and $f_{f}$ as Data;
  6    l.add(r,q);
  7   **end**;
  8   sort *l*, quantity of available attendants;
  9   l← execute *Human Resources Deallocation Algorithm* using *l* and allocated attendants of *v* as Data;
10   sort *l*, quantity of available attendants;
11   **forall**
*Room r on list l*
**do**
12    $l_{r}\leftarrow$ sort *l*, quantity of available attendants with room *r* specialty;
13    $available_{r}\leftarrow$ list of all human resources available for allocation with room *r* specialty;
14    execute *Human Resources List Scheduling Algorithm* using *r* and $l_{r}$ as Data;
15    **if**
*r need more attendants*
**then**
16        Execute *Human Resources Allocation Algorithm* using *r*, $l_{r}$ and $available_{r}$ as Data;
17    **end**
18   **end**
19   h← rooms of *l* vector;
20   **return**
*h*;
21 **end**


In order to achieve a balanced reallocation of human resources, we developed a variation of the dynamic List Scheduling algorithm [49], which was originally used for process scheduling. Here, all hospital rooms are in a list ordered by the number of attendants available for reallocation. In that way, whenever a room *r* needs more attendants, the elasticity manager checks for available attendants, with room *r* specialty, in the first room of the list. If attendants are available, then they are reallocated to the room lacking them, and the list is sorted again. If more attendants are needed, the algorithm checks the first room in the list again, and so forth, until the room obtains all the required attendants.

Figure 11 illustrates the reallocation process, where Room 1 needs three more attendants and Rooms 2 and 4 have some free attendants. Following the logic of the adapted List Scheduling algorithm, in the first round, Room 2 is the first in the list, with three available attendants, and gives an attendant for Room 1. In the second round, even though all rooms in the list have the same number of free attendants, Room 2 remains at the top of the list, so another attendant is reallocated. Finally, in the third round, Room 4 becomes the first on the list, since it has two free human resources (as opposed to Room 2, which has only one), and an attendant of Room 4 is reallocated to Room 1.

As in the reactive strategy, in proactive elasticity, we employ a cooldown-based strategy to prevent hysteresis of human resources. If the reallocation process is not enough to improve the attendance level of the hospital, ElHealth proposes the allocation of new human resources to hospital care. Lastly, if ElHealth identifies that the future demand for care of all hospital rooms is very low and that the deallocation of attendants of some room does not harm the whole, ElHealth proposes the deallocation of attendants allocated outside of their regular working hours.

## 4. Evaluation Methodology

We assess the performance of ElHealth through simulations in a virtual hospital environment. Considering the unavailability of data, the hospital environment was defined based on synthetic workloads. These data and its parameters are detailed in Section 4.2. According to [50], synthetic workloads can be considered a representative form to evaluate elasticity in computational clouds. ElHealth was implemented mainly in Java, except for the ARIMA method, which was implemented in Python. For hospital queues simulation, we used a clock with discrete increments of ten seconds. At each advance in the simulation clock, our simulator verifies the patients who are in care and those who should leave the care. At each monitoring cycle, the arrival of patients should be checked. The data probability distributions were generated using triangular distributions (more details in Section 4.2), as implemented by *StdRandom* [51].

### 4.1. Considered Scenarios

Given the hospital simulation procedure, we consider three different scenarios for analysis. In all scenarios, we used the same input parameters. The differences in the scenarios are related to the use of the proposed model in the hospital environment and will be described as follows:**S1:** **Hospital without ElHealth:** in order to have data for comparison, the first test scenario is based on the simulation of a non-elastic hospital**S2:** **Smart hospital with ElHealth’s reactive elasticity:** the second scenario focuses on the simulation of the hospital environment with the use of the allocation, reallocation, and deallocation of human resources proposed in the ElHealth model, using reactive elasticity approach.**S3:** **Smart hospital with ElHealth’s proactive elasticity:** the third scenario is similar to the second, based on the simulation of the hospital environment with ElHealth’s elasticity model, but unlike the previous scenario, using proactive elasticity approach.

### 4.2. Performance Evaluation Parameters

To perform the simulation of the hospital environment, we use the data collected in the study of Capocci et al. [25] performed in a hospital environment located in Guarulhos City, in the state of São Paulo in Brazil. According to Capocci et al. [25], all patients upon entering the unit first go through reception, where a Personal Health Record (PHR) [52] is prepared. After this preparation, patients are referred to waiting for triage. In the triage procedure, the patients are examined by the nursing team and classified into priorities according to the urgency of the health problem and are referred to waiting for medical attention. In polyclinic analyzed by Capocci et al. [25], after first medical attention, 24% of patients are referred for x-ray exam, 37% for laboratory examinations (blood test, for example), 8% for electrocardiograms (ECG) exam, and 31% do not need more than physician examination. Also after doctor treatment room, only 1% of patients do not take medication and are released with only one prescription, but 50% of patients require intravenous medication, 30% intramuscular injection and 19% inhalation medication. After the exams, 60% of patients need to return to the doctor, and 40% are released. After a return care, 78% of patients are released, 2% need new exams, and 20% require new medication.

Also, according to Capocci et al. [25], the care time in each room of the hospital environment follows a triangular distribution, with minimum and maximum times and a more frequent average time. Table 4 shows the distributions for all possible care in this hospital unit, as identified by [25] in their study. All other parameters used in our simulation can be found in [25].

In Brazil, the working model adopted for hospital environments is the so-called 12 × 36 h. According to Brazilian Law No. 13,467 [53], under this work regime, an employee can work for twelve consecutive hours (with a one-hour pause for lunch) and must rest for 36 h before a new work shift of 12 h starts. Under this regime, four health professions alternating shifts is enough to ensure a single position for 24 h, seven days a week. Also, according to the understanding of the law, if for any reason an employee needs to work within their rest period, it should be treated as overtime, unless the hours are compensated at another time. Thus, while a human resource of the hospital is in working time, three other employees who perform the same function are in their paid-rest period. According to Brazilian Decree-Law No. 5,452 [54], the minimum rest period between two working days must be eleven consecutive hours. In that way, even if there are overtime hours, an employee must rest eleven hours to return to the next work shift. Thus, these three resting employees shall not be arbitrarily available to a new allocation. In particular, any resting employee is only available under the following rules:**Rule** **1:**The minimum rest period for a human resource to be available for allocation is eleven hours;**Rule** **2:**An allocated employee cannot works outside of the regular work shift for a long time period. The largest possible work period allowed in Brazilian legislation is twelve hours. Thus, an allocated employee cannot work more than twelve hours;**Rule** **3:**Allocated employees must be deallocated no later than 11 h before they next normal work shift; and**Rule** **4:**Each employee must meet one of the 36 h rest periods within the same week in order to comply with a law determination that requires all workers to have a 24 h paid-rest period per week.

As our case study is based on Brazilian hospital data, we have set thresholds appropriate to our reality. So, based in Brazilian Law Project of 14 June 2018 [55] that proposes a maximum waiting time for care in hospitals, clinics, and laboratories of 30 min on regular days (from Monday to Sunday), we define ElHealth’s maximum load (i.e., 100%) in 30 min. Based on several works [12,47,56,57], we are using 4 combinations of thresholds when evaluating the second scenario, so considering 30% (9 min) and 50% (15 min) for lower threshold, and considering 70% (21 min) and 90% (27 min) for upper threshold. For proactive elasticity, we set ElHealth’s upper threshold in 30 min, (i.e., maximum load previously defined), and we set ElHealth’s lower threshold in 9 min (30% of maximum waiting time). For elasticity actions, we set 10 min for reallocation process (human resources movement between rooms), and 60 min for allocation process (to simulate the movement of a new human resource to the hospital).

### 4.3. Workload

We use the human resources allocation found in [25] research, where 11 health professionals were allocated, 24 h a day, seven days a week, through more than one work shift. To be specific, health professionals were allocated as follows: 2 attendants in a reception; 1 nurse working in patient triage; 2 doctors acting in doctors treatment rooms; 2 nurses working with collection exams; 2 nurses working throughout the medication area; 1 nurse acting on the electrocardiogram; and 1 radiology technician acting with the X-ray exams.

Regarding patients load, we modeled four workloads: constant, ascending, descending, and wave. The idea of using different load behaviors for the same application is used to observe how the input load can impact saturation points, bottlenecks, and the addition or removal of resources [56]. These four behaviors of workload are based on those proposed by [56]. Besides these four loads are representative to evaluate elasticity, wave workload is the most closely to the hospital reality, and the ascending workload represents the behavior of the model in a situation of increased patient load, which could be caused, for example, by a viral outbreak or epidemics. We want to emphasize that the ascending workload demonstrates the worst possible case, with an increasing entry of patients into a hospital. Figure 12 presents a representation of each workload of the model. The *x* axis expresses the time available in one day of care in the hospital unit, while the *y* axis represents the arrival of patients at each instant of time.

Since the workloads generate decimal numbers, we established a strategy to generate integers for the arrival of the patients in the hospital environment. This occurs because, in a real environment, it is not possible the arrival of 0.2 patients or 1.7 patients, for example. Thus, we adopted a load accumulation strategy, where if at any given moment there is something between 0.1 and 0.9 patient, this value is accumulated with next instant load. An example would be an instant with a load of 0.6 patient. Since there would not be an integer charge, a patient would not be introduced into the system, and the charge would accumulate for the next instant of time. At the next moment, with a new load of 0.6 patient, the accumulated load would be 1.2 patient, resulting in the entry of 1 patient in the hospital. Thus, there would be still 0.2 patient, which would be accumulated for the next instant and so on.

### 4.4. Performance Evaluation Metrics

In order to evaluate the proposed model, the following metrics are considered:Maximum waiting time for care;Human resources cost;Elastic number of human resources used.

To evaluate the waiting time, we used as parameter the variation of the maximum waiting time between the scenarios and the adequacy of the maximum waiting time to the established limits. To determine the human resources cost, we had to propose a way to measure the cost of a human resource in normal working hours and the cost of a human resource outside of its working hours. According to Brazilian Law No. 13,467 [53] and Brazilian Decree-Law No. 5452 [54], the overtime pay will be at least 50% (fifty percent) higher than the normal hour. In this way, a health professional allocated outside of its working shift costs 50% more than an employee during its working shift. Based on this, we devised Equation (Equation 7) for Human resources cost as follows:(7)$$Cost\left(t_{i},t_{f}\right)=\frac{1}{t_{f}-t_{i}}\sum_{t_{n}=t_{i}}^{t_{f}-1}HR\left(t_{n}\right)+\left(1.5\cdot AllocatedHR\left(t_{n}\right)\right)$$
where $HR(t_{n})$ refers to all human resources in their working shift at $t_{n}$ time instant, and $AllocatedHR(t_{n})$ refers to all allocated, or in the process of allocation, human resources outside their regular working hours at $t_{n}$ time instant. With regard to human resources number, we proposed a metric for compare elastic and non-elastic health professional allocation, where we expect that our model uses the existing health professionals in the hospital in an optimized way. Thus, static allocation of S1, with eleven employees working, can be compared to ElHealth elastic allocation, with the number of human resources varying throughout the day. Table 5 presents all the evaluation metrics described above, relating the results expected for the second and third scenario with the use of ElHealth, when compared to the current hospital environment, without the ElHealth model.

## 5. Performance Evaluation and Results Analysis

Based on the evaluation methodology proposed for the ElHealth model, we performed twelve simulations of the proposed hospital environment in order to collect results for analysis. For each proposed scenario, between S1, S2, and S3, a simulation was performed for each of the workloads, constant, ascending, descending, and wave.

For the maximum waiting time metric, we expected a decrease in patients’ waiting for care. Figure 13 shows the maximum waiting time identified for each workload in the proposed scenarios over the simulated one-week period. We perceive a significant reduction in the maximum waiting time between S1 and S2, and a second diminution when comparing S2 and S3, regardless of the workload used. After a thorough analysis, we can identify that in S3 for reception, triage, doctor treatment, and collection exams rooms, at no time was measured waiting time longer than 30 min, regardless of the workload used. As for medication, X-ray, and electrocardiogram rooms, there were a few moments when this limit was exceeded. Through the collected data, we identify a significant reduction in waiting time with the use of the reactive and proactive elasticity approaches for human resources organization when compared to the hospital without the use of the elasticity. Thanks to the reactive procedures, ElHealth has shown to decrease the waiting time by 96.13%, 95.27%, 96.05% and 93.4% for constant, ascending, descending, and wave workloads, respectively, as compared to the scenario where no human resources reorganizations are performed. In proactive procedures, ElHealth has shown to decrease the waiting time by 96.66%, 96.73%, 97.06% and 96.65% for constant, ascending, descending and wave workloads, respectively, as compared to the non-elastic hospital.

For human resources cost metric, we expected an increase in the cost between scenarios. Figure 14 presents the human resources cost for each workload in S2 and S2, the scenarios where the cost can variate. We can observe that cost ranged from 11 to 17.77 per hour, in the reactive approach, and ranged from 11 to 18.47 in the proactive approach. Furthermore, as exposed in the aforementioned Figure 14, whenever ElHealth costs increase, the patients’ waiting time decreases. We can also see that the proactive approach achieved the most significant reduction in waiting time, with more cost than the reactive approach. In reactive procedures, the cost increased by 0.64%, 5%, 13.27% and 9.27% for constant, ascending, descending, and wave workloads, respectively. In proactive procedures, the cost increased by 7.09%, 7.36%, 22.82% and 3.27% for constant, ascending, descending, and wave workloads, respectively.

For the elastic number of human resources used metric, we expected an increase in the number of professionals in the hospital, as well as a variation of this number over the hospital care period. Figure 15 presents the elastic number of human resources used for hospital care in S3, the only scenario where the number of employees can variate. We can observe that the elastic number of human resources ranged from 11 to 14 per hour. Although there are moments with the allocation of up to 14 health professionals in care, the average per hour of care professionals turns out to be slightly lower depending on the time it takes for an employee to be allocated or reallocated in the hospital. Furthermore, as exposed in the aforementioned Figure 15, whenever ElHealth reallocates or allocates peoples for care, the patients’ waiting time decreases.

### Discussion

Based on established metrics, we can note that the ElHealth model was able to improve the performance of the simulated hospital environment in all workloads used. Table 6 presents all the results found in each of the proposed evaluation metrics, highlighting the best results in green and the worst in red. As proposed in our evaluation methodology, we expected that the maximum waiting time presented a gradual decrease between scenarios S1, S2, and S3, and this in fact occurred, fulfilling the objective of this metric. For human resources cost, we expected an increase between scenarios S2 and S3, and our model has met expectations. For the elastic number of human resources used, an increase in the result was expected between scenarios S2 and S3, and our model once again was able to meet the proposed goal. Thus, the expected results in the evaluation methodology were achieved through the use of the ElHealth model in the proposed hospital environment.

For maximum waiting time metric, our objective was the time reduction. As already shown, the ElHealth model was able to reduce the waiting time for the proposed hospital environment significantly. However, although the average maximum waiting times for the S3 scenario were within the established limit (9.42 min with constant workload, 12.7 min with ascending workload, 15.65 min with descending workload and 12.8 with wave workload), when we analyzed the longer waiting time identified in all the simulation period, the upper limit was exceeded (39, 48, 86 and 70 min with constant, ascending, descending and wave workloads, respectively). We believe that this occurred due to the limitations of the hospital environment used as the basis for this simulation. As there were not many care stations available to be allocated new human resources, our model was not able to reach the goal in this hospital environment. For human resources cost metric, we expected an increase among the proposed scenarios, and that is precisely what happened. When we compare with the previous metric, the increase in human resources cost is inversely proportional to the waiting time decrease. In the reactive approach, we had a considerable improvement in waiting time reduction, with little increase in cost. For proactive elasticity, we have a new improvement in waiting time reduction, with a new increase in the cost. Although the proactive approach has a higher cost, we believe it is still more efficient than the reactive approach because it can further decrease waiting time for medical care, anticipating more potential health problems. For the elastic number of human resources used metric, we expected an increase in the average number of human resources between scenarios C2 and C3, and this also actually occurred.

## 6. Conclusions and Future Work

IoT sensors allow smart hospitals capable of tracking people and objects in real-time. With this data, computer systems can be used to generate knowledge and value for hospital managers. This work puts efforts in this direction, taking data captured from IoT sensors and generating decision-making value on them. Thus, this article presented the ElHealth model. Unlike related work, ElHealth not only proposes the use of elasticity to anticipate eventual problems in the future but also presents a model to allocate, migrate and deallocate people in hospitals in such a way to provide benefits at patients viewpoint. Using IoT-sensors and an ARIMA-based prediction engine, we can instrument a smart hospital to collect data in time-series, so better arranging professionals and either preventing or mitigating patient treatment problems, which sometimes are related to life or death issues. In this way, we extended the concept of elasticity from cloud computing to the context of human resources management, while proposing new mathematical formalisms, algorithms, and definitions to provide a dynamic and elastic allocation of professionals in hospital environments.

We expect that the model proposed in this work can help to decrease the waiting time of patients for healthcare. The idea is to provide such facility in a transparent way for the patients, i.e., they do not need to follow additional procedures in the hospital, but only wear a wristband which serves as identification. We also hope to, with the use of ElHealth, we can identify bottlenecks in the patients care flow and help optimize processes in healthcare environments. Moreover, the provided data can also be used for decision making in terms of changes in hospital capacity and infrastructure. In ElHealth’s case study, the waiting time is decreased by 96.4% and 96.73% for reactive and proactive approaches, respectively. In the reactive approach, we had a considerable improvement in terms of waiting time reduction, with little cost increasing. On the other hand, with the proactive approach, we had more waiting time reduction, with an increase in the cost. Even with the higher cost, we believe that proactive elasticity is more efficient than the reactive approach since with a shorter waiting time, more potential health problems can be anticipated.

Although presenting encouraging results, we envisage some limitations that must be addressed on implementing ElHealth model in a real hospital environment: (i) employees and patients must carry their identification tags throughout their time in the smart hospital; (ii) ElHealth only generates notifications for human resource, but the effective movement of staff in hospital environments depends on their individual decision to follow the recommended guidance; (iii) previous installation of RTLS sensors in corridors and doors of the hospital.

As future work, we envisage the implementation of the IoT system, as well as the development of a prototype that implements all the modules and algorithms proposed by ElHealth, so enabling the deploying in a real hospital environment. Another possibility concerns the adaptation of the model to use other prediction algorithms on the proactive approach, including Artificial Neural Networks and Random Forest approaches. Also, we visualize a new approach to perform an evaluation based on a function incorporating constant, ascending, descending and wave workloads with different coefficients, since in an actual hospital environment a mix of these workloads could appear and modify over time.

## Figures and Tables

**Figure 1 sensors-19-03800-f001:**
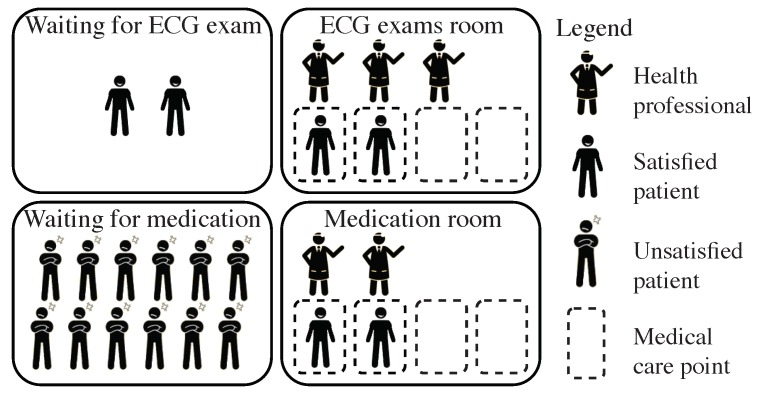
Example scenario where there are more attendants examining than medicating patients, even though the number of patients waiting for exams is considerably smaller than those waiting to receive some medication, generating dissatisfaction for patients awaiting medication.

**Figure 2 sensors-19-03800-f002:**
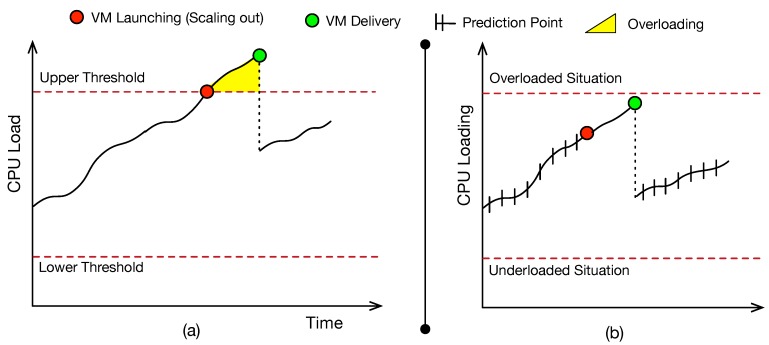
Elasticity approaches: (**a**) reactive; (**b**) proactive.

**Figure 3 sensors-19-03800-f003:**
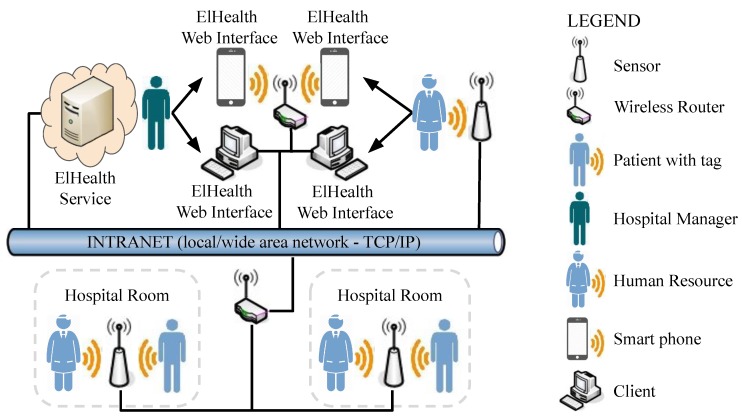
Components and network view in ElHealth model with (**i**) ElHealth Web Interface; (**ii**) ElHealth Service, for information processing and decision making; (**iii**) a RTLS, for track users’ tags; and (**iv**) Hospital managers, patients, or human resources.

**Figure 4 sensors-19-03800-f004:**
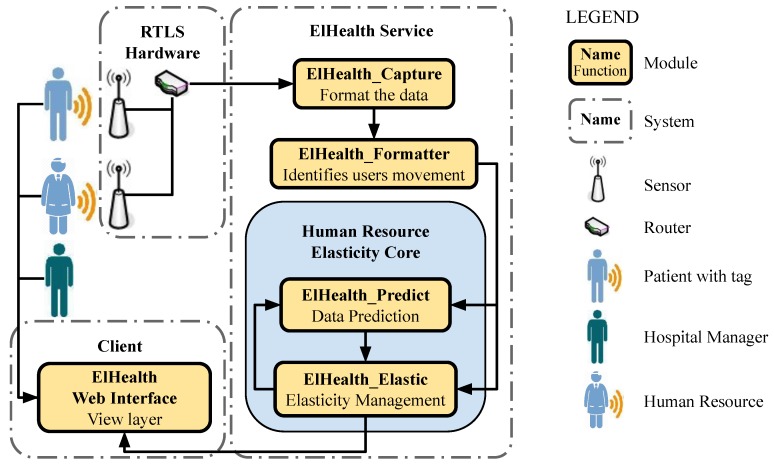
ElHealth model architecture detail where the information flow starts in ElHealth_Capture module that receives users’ movement records from RTLS sensors, and goes through different handlings over proposed modules, until the exhibition of elasticity notifications in ElHealth Web Pages.

**Figure 5 sensors-19-03800-f005:**
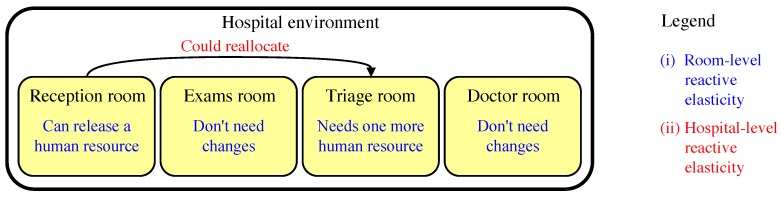
Multi-level Reactive Elasticity of Human Resources example with (**i**) room-level reactive elasticity, and (**ii**) hospital-level reactive elasticity.

**Figure 6 sensors-19-03800-f006:**
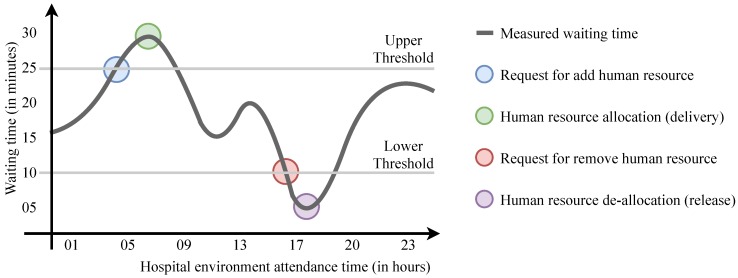
Reactive elasticity example based on waiting time for care adopted by ElHealth, where the delivery and release of human resources occur after the thresholds are reached.

**Figure 7 sensors-19-03800-f007:**
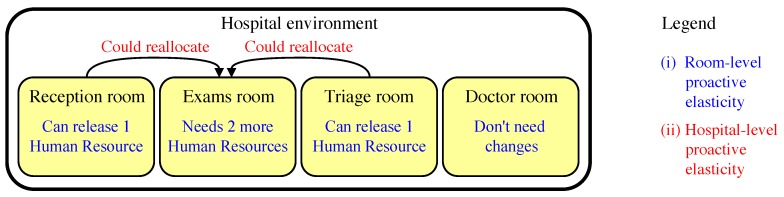
Multi-level Proactive Elasticity of Human Resources example with (**i**) room-level proactive elasticity, and (**ii**) hospital-level proactive elasticity.

**Figure 8 sensors-19-03800-f008:**
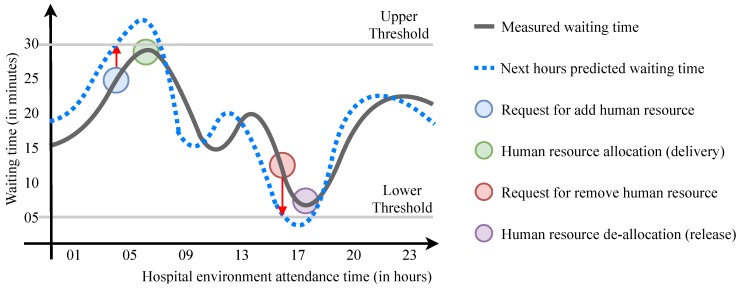
Proactive elasticity based on predicted waiting time for care adopted by ElHealth, where the delivery and release of human resources occur before the thresholds are reached.

**Figure 9 sensors-19-03800-f009:**
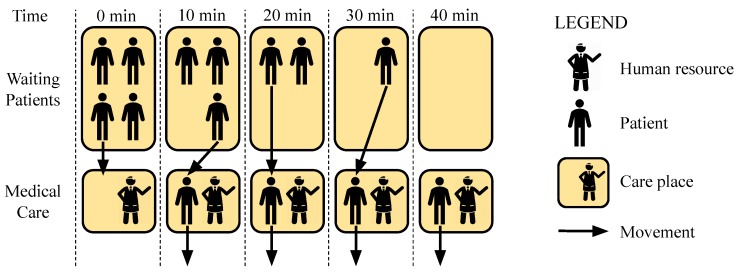
Calculating ECT in a hospital room $r_{1}$ with 4 patients waiting, and average care time of 10 min. In this hypothetical situation, at 0 min instant the first patient was called to the care. In 10 min instant, the first patient ends their care and goes away, so the second patient is designated to care, and so on, until instant 40 min, when the last patient is released. Thereby, all patients are attended within 40 min. Applying Equation (Equation 4), we obtain $ECT\left(r_{1},t_{i},t_{f}\right)=ACT\left(r_{1},t_{i},t_{f}\right)\cdot ENP\left(r_{1},t_{i},t_{f}\right)=10\times 4=40$ min.

**Figure 10 sensors-19-03800-f010:**
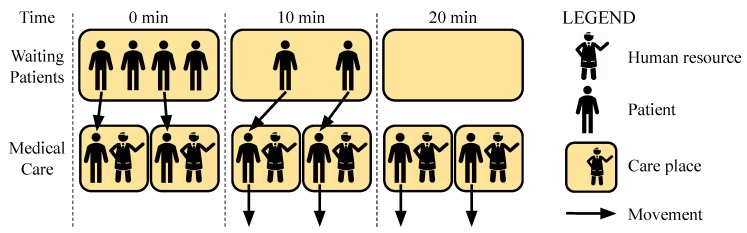
Calculating the ECT in a hospital room using parallel allocation of attendants, with 4 patients waiting, average care time of 10 min, and 2 attendants. In this hypothetical situation, at 0 min time instant, there were 4 patients waiting and none in attendance by doctors, so the first two patients were called to care. In 10 min instant, the first two patients are released, and the last two patients are designated to care. Thus, at 20 min instant, the last two patients are released. Thereby, all patients are attended in only 20 min. Using Equation (Equation 5), we obtain: $HRES\left(r_{1},t_{i},t_{f}\right)=\frac{ECT(r_{1},t_{i},t_{f})}{ANA(r_{1},t_{i},t_{f})}=\frac{ACT\left(r_{1},t_{i},t_{f}\right)\cdot ENP\left(r_{1},t_{i},t_{f}\right)}{ANA(r_{1},t_{i},t_{f})}=\frac{10\times 4}{2}=20$ min.

**Figure 11 sensors-19-03800-f011:**
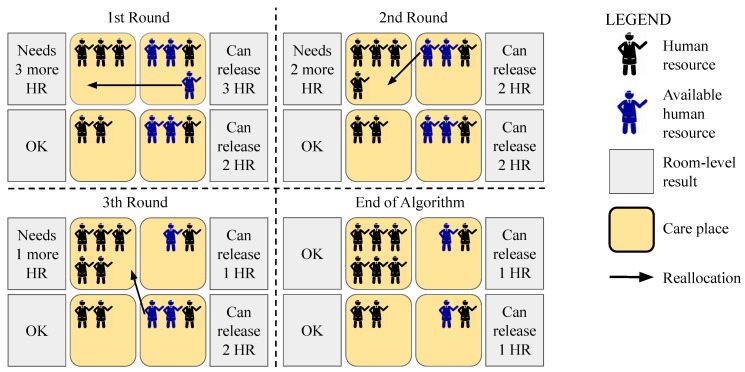
Reallocation through the adapted List Scheduling algorithm, with a sorted list of 4 rooms, and 12 attendants, where Room 1 needs to allocate more 3 attendants.

**Figure 12 sensors-19-03800-f012:**
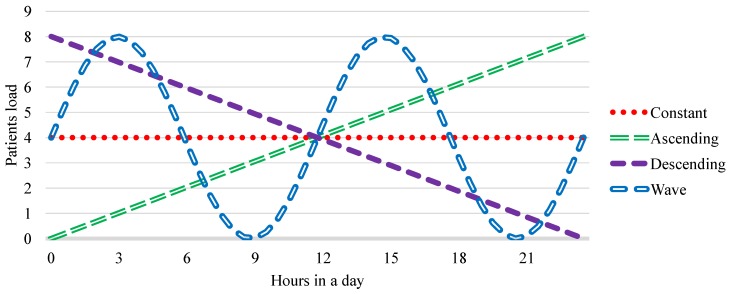
Graphical representation of workloads used in ElHealth tests, where *x* axis expresses time available in one day of care, while *y* axis represents the arrival of patients at each time instant.

**Figure 13 sensors-19-03800-f013:**
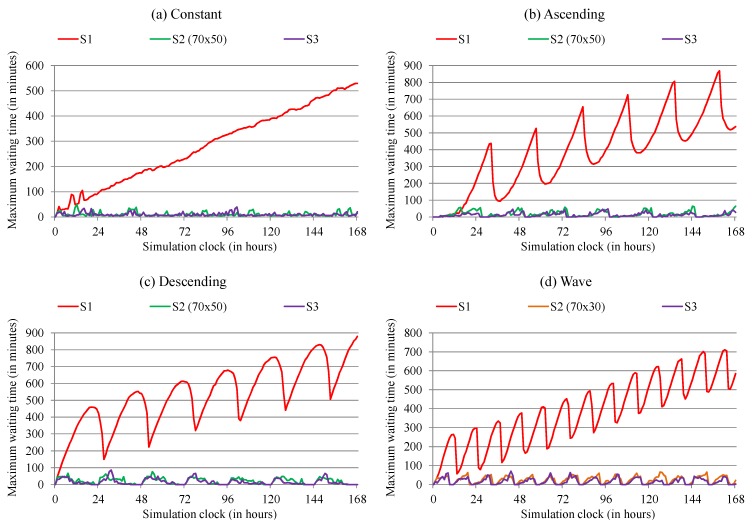
Maximum waiting time at the hospital for each of the proposed scenarios, S1 (in red), best result between thresholds for S2 (in green for 70 × 50 and in orange for 70 × 30), and S3 (in purple), using (**a**) constant, (**b**) ascending, (**c**) descending and (**d**) wave workloads.

**Figure 14 sensors-19-03800-f014:**
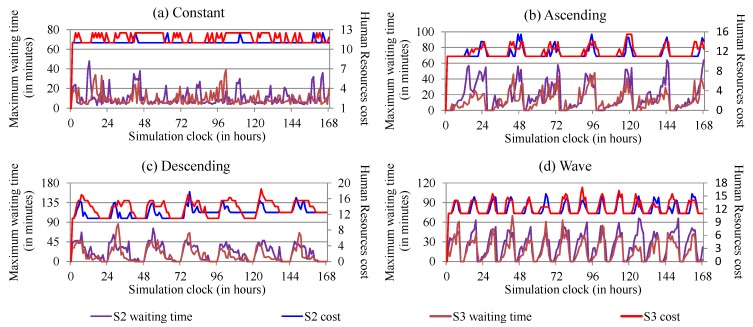
Human resources cost compared with maximum waiting time at the hospital using (**a**) constant, (**b**) ascending, (**c**) descending and (**d**) wave workloads in S2 (best result between thresholds) and S3.

**Figure 15 sensors-19-03800-f015:**
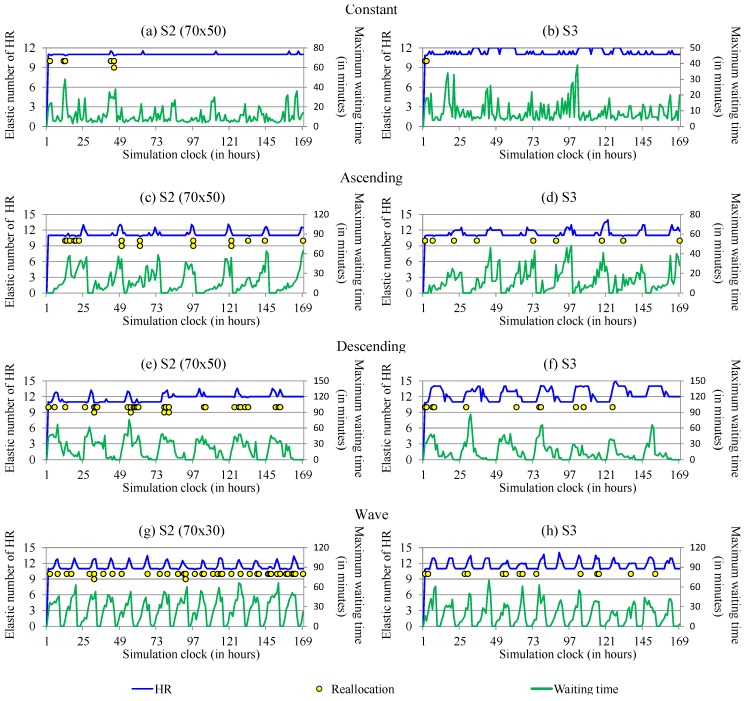
Elastic number of human resources used compared with maximum waiting time at the hospital for (**a**) constant workload in best result between thresholds for S2, (**b**) constant workload in S3, (**c**) ascending workload in best result for S2, (**d**) ascending workload in S3, (**e**) descending workload in best result for S2, (**f**) descending workload in S3, (**g**) wave workload in best result for S2 and (**h**) wave workload in S3.

**Table 1 sensors-19-03800-t001:** Reactive and proactive related work comparison.

Work	Focus	Elasticity	Prediction Algorithm
Al-Dhuraibi et al. [18]	Cloud applications	Reactive	–
Elastic-RAN [19]	C-RANs	Reactive	–
ElCity [12]	City energy	Reactive	–
Hanafy et al. [20]	Cloud applications	Proactive	Time-series (ARMA)
Proliot [21]	IoT applications	Proactive	Time-series (ARIMA and WMA)

**Table 2 sensors-19-03800-t002:** Human resources in healthcare related work comparison.

Work	Focus	Proposed Solution	Data Prediction Model	Human Resources Defi-Ciency
Capocci et al. [25]	Improve patient flow, decreasing the waiting time for care	Identify bottlenecks to propose human resources movement	Uses Queueing theory to estimate patient’s arrivals.	Proposes a nurse reallocation based on waiting time for screening process
Vieira and Hollmén [26]	Deficiency of resources to perform patients’ care	Identify the resources needed to ensure the patient’s care flow	Uses Nearest Neighbours and Random Forest to predict future resources usage	Only provides data to support decision- making
Ishikawa et al. [35]	Deficiency of doctors for current patients’ care demand in Japan (Hokkaido)	Identify health doctors distribution and suffi- ciency to propose ways for guarantee care for demand	Uses System Dynamics (SD) and Geographic Information System (GIS) to predict distribution and sufficiency of doctors	Proposes a plan for training physicians that considers geographic requirements
Liu et al. [36]	Deficiency of doctors for current patients’ care demand in global scale	Identify health doctors distribution and suffi- ciency until 2030 in order to compare with demand projections	Uses an economic model and a Generalized Li- near Model to predict distribution and sufficiency of health professionals, and patients’ demand	Only provides data to show the problem escalation, to support solutions proposal by the international community
Graham et al. [27]	Emergency depart- ments crowding and the negative conse- quences for patients	Use of data mining using machine learn- ing techniques to predict admissions in a hospital	Uses logistic regression, decision trees and Gradient Boosted Machines to predicts patients’ arrival in emergency	Only provides data to support decision- making of hospital managers

**Table 3 sensors-19-03800-t003:** Mathematical notation of ElHealth.

Nomenclature	Description
*r*	Hospital room
$t_{i}$	Initial time instant
$f_{i}$	Future initial time instant
*a*	Allocated attendants
size(x)	Size of a *x* vector
$ACT(r,t_{i},t_{f})$	Average Care Time
$ANA(r,t_{i},t_{f})$	Average Number of Attendants
$NIP(r,t_{n})$	Number of Incoming Patients
$ECT(r,t_{i},t_{f})$	Estimated Care Time
$HRES(r,t_{i},t_{f})$	Human Resources Elastic Speedup
$t_{n}$	Specific *n* time instant
$t_{f}$	Final time instant
$f_{f}$	Future final time instant
$CV(r,t_{i},t_{f})$	Care Vector
CDT(x[i])	Care Duration Time
$NA(r,t_{n})$	Number of Attendants
$NWP(r,t_{i})$	Number of Waiting Patients
$ENP(r,t_{i},t_{f})$	Estimated Number of Patients
$PHRES(r,a,f_{i},f_{f})$	Proactive Human Resources Elastic
	Speedup

**Table 4 sensors-19-03800-t004:** Triangular distributions of probability for care times.

Attendance	Attendance Time		
	Lower	Mode	Upper
**Reception room**			
PHR preparation	2 min	3 min	5 min
**X-Ray exams room**			
X-Ray exam	10 min	15 min	23 min
**Medication room**			
Intramuscular injection	3 min	3.5 min	5 min
Intravenous and inhala-	0.5 min	1.5 min	2.5 min
tion preparation			
Intravenous medication	40 min	70 min	120 min
Inhalation medication	8 min	10 min	13 min
**Triage room**			
Triage process	5 min	8 min	10 min
**Doctor treatment room**			
First care with doctor	5 min	11 min	16 min
Return care with doctor	4 min	7 min	10 min
**Collection exams room**			
Laboratory exams	6 min	8 min	13 min
**Electrocardiogram exams room**			
ECG exam	30 min	45 min	60 min

**Table 5 sensors-19-03800-t005:** Evaluation metrics and expected results in each scenario.

Scenario	Maximum Waiting Time	Human Resources Cost	Elastic Number of Human Resources Used
S1	Current	Current	11 by work shift
S2 (Expected)	Less than S1	More than S1	11 or more by work shift
S3 (Expected)	Less than S2	More than S2	11 or more by work shift

**Table 6 sensors-19-03800-t006:** Evaluation metrics and results found in each of the proposed scenarios, using constant, ascending, descending and wave workloads, where the best results for each metric are highlighted in green and the worst in red.

Workload	Scenario	Thresholds		Maximum Waiting Time (in Minutes)		Human Resources Cost	Elastic Number of Human Resources
		Upper	Lower	Average	Upper		
Constant	S1	-	-	282.32 (±147.7)	529	11	11
	S2	90	50	21.71 (±15.8)	58	11.04	11.01
		70	50	10.93 (± 8.6)	48	11.07	11.02
		90	30	19.67 (±14.8)	57	11.01	10.99
		70	30	15.22 (±11.1)	64	11.07	10.99
	S3	-	-	9.42 (± 6.7)	39	11.78	11.33
Ascending	S1	-	-	388.81 (±215.8)	868	11	11
	S2	90	50	27.28 (±21.5)	86	11.60	11.22
		70	50	18.39 (±17.3)	64	11.55	11.19
		90	30	28.82 (±19.8)	66	11.70	11.23
		70	30	20.14 (±19.1)	88	11.57	11.19
	S3	-	-	12.70 (±11.7)	48	11.81	11.36
Descending	S1	-	-	532.01 (±182.0)	880	11	11
	S2	90	50	24.99 (±23.4)	87	11.62	11.23
		70	50	21.01 (±18.2)	76	12.46	11.75
		90	30	28.23 (±25.0)	97	11.68	11.25
		70	30	23.05 (±23.0)	83	11.86	11.34
	S3	-	-	15.65 (±17.6)	86	13.51	12.42
Wave	S1	-	-	384.18 (±171.7)	711	11	11
	S2	90	50	33.57 (±24.3)	92	11.87	11.33
		70	50	28.99 (±22.6)	95	12.08	11.42
		90	30	34.68 (±22.5)	75	11.89	11.33
		70	30	25.37 (±19.1)	66	12.02	11.38
	S3	-	-	12.88 (±12.4)	70	11.36	11.59

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
