# Peer review of "Towards Evaluating Proactive and Reactive Approaches on Reorganizing Human Resources in IoT-Based Smart Hospitals"

_sensors, 2019, doi:10.3390/s19173800_

Round 1

Reviewer 1 Report

The paper presents an evaluating proactive and reactive approach on reorganizing human resources in IoT-based smart hospitals. The paper is interesting and well-written. It is easy to follow the whole paper. The paper is suitable for both expert and non-expert readers. The paper is well structured.

I have a few minor comments. There are some typos in the papers please go through the whole paper and polish it.

It is more persuasive if the proposed system can be verified in actual hospitals. Since actual hospital has many exceptional cases. For instance, in some emergencies cases , most of valuable resources (i.e., experiences doctors and the whole operation group) are utilized. As a results, resources are not available. How the proposed system can deal with the dynamic cases or unexpected emergent cases.

In general, the paper has some merits. It is good for the community

Author Response

The point-by-point response to the reviewer’s comments can be found in attachments.

Reviewer 2 Report

This paper investigates strategies for optimizing the allocation of human resources (doctors, nurses, technical staff, …) in hospitals and medical clinics, in order to minimize the patient waiting time. This is a very important topic since in some cases the patients can have damage or even life threats while waiting for his/her turn (f.i. emergency rooms).

Specifically, exploiting an IoT-based infrastructure enabling tacking of patients and health professionals (f.i. tagged by RFID bracelet or badge) authors propose a model i.e. Elastic allocation of human resources in smart healthcare environments (ElHealth) - to monitor occupancy of the hospital rooms and optimize patient care demand.

To validate the proposed model simulation techniques have been applied, taking data input from a study collecting and sharing a clinical dataset related to the patient hospitalization cycle.

Three scenarios have been compared: 1) classical hospitals (without any IoT infrastructure), Smart hospitals exploiting the ElHealth 2) reactive or 3) proactive elasticity model.

Concerning patients four workloads have been analyzed: constant, ascending, descending, and wave.

Regardless of the workload the proposed model reduce the patient waiting time. However, as expected, results show that cost increases when increasing efficiency. The proactive model showed the greater waiting time reduction, with more cost vs the reactive algorithm.

This paper is clear, well organized and easy to read. The related work is adequate discussed and include recent research results. An additional reference can be introduced: Berg, B., Longley, G., & Dunitz, J. (2019). Improving clinic operational efficiency and utilization with RTLS. Journal of medical systems43(3), 56.

The proposed model is interesting. The method is solid and exploits consolidated knowledge. In my opinion, examples sound a little bit simple. The analysis of results is accurate and can be useful for other researchers.

Few suggestions:

In Figure 3, please to substitute “local/wide area network - IP (Internet Protocol)” with “local/wide area network – TCP/IP” in order to include the TCP (Transfer Control Protocol) which is necessary to guarantee reliability to the Internet Protocol… and I also suggest to change “web page” with “web interface” since authors refer to a web system/service (not to a website). The advantage of a web application is to have device-independent web interfaces on smartphone, tablet or computer...

Concerning patients the four workloads have been analyzed separately, but in an actual hospital environment a mix of these workloads could appear and modify over time, so an idea could be to generalize it as a function incorporating of all these four components with different coefficients (that can be modulated as parameters).

Please notice that people with color blindness might have problem in interpreting graph based on color… maybe different line patterns could help the graph visual interpretation…

Minors

In Figure 13, Wave workload, S2 (70x30) is in red/orange color while in other workloads is in green.

Please check for missing spaces, for instance:

Fig 9     minutes.In

pag 15   attendants.In

Author Response

(The authors gave the same response as above.)

Round 2

Reviewer 2 Report

In my opinion, this study is now ready to be published. Authors applied all changes required by reviewers, thus this paper new version is more accurate and clarifies some aspects.

This paper investigates strategies for optimizing the allocation of human resources in hospitals to minimize the patient waiting time. This is a very important topic and critical in an emergency context. I believe this research can help other researchers to enhance knowledge in this field.